# Effects of aging and exercise habits on blood flow profile of the ocular circulation

Chihyun Liu[1], Tatsuhiko Kobayashi[2], Tomoaki Shiba[3], Naoyuki Hayashi[1,4]*

1 School of Environment and Society, Department of Social and Human Sciences, Tokyo Institute of Technology, Tokyo, Japan, 2 Department of Ophthalmology, School of Medicine Toho University, Tokyo, Japan, 3 Department of Ophthalmology, International University of Health and Welfare, Narita Hospital, Chiba, Japan, 4 Faculty of Sport Sciences, Waseda University, Saitama, Japan

* naohayashi@waseda.jp

**Data Availability Statement:** No data can be open since it was not allowed in the IRB process due to privacy policy, since data contain potentially identifying patient information. The Research

## Abstract

### Purpose

We examined the effects of aging and exercise habits on the ocular blood flow (OBF) and its profiles throughout the optic nerve head region and choroidal area. We hypothesized that exercise habits reduce the stiffness of vessels in the ocular circulation, which generally increases with aging.

### Methods

Participants in a medical checkup program (698 males and 192 females aged 28 to 80 years) were categorized into 2 groups (with and without exercise habits) based on participant self-reporting and the definition of the Ministry of Health, Labor and Welfare of Japan (MHLW). OBF in the right eye was measured and analyzed using laser speckle flowgraphy. The blow-out time (BOT), which is the time during which the blood flow is higher than half of the mean of the minimum and maximum signals during one heartbeat, was calculated as an index of the blood flow profile. BOT has been used as an indicator of the flexibility of blood vessels.

### Results

BOT significantly decreased with aging. Neither the self-reported nor MHLW-based exercise habits significantly affected the ocular circulation.

### Conclusion

These results indicate that the stiffness of the ocular vessels increases with aging, and this cannot be prevented by exercise habits.

## Introduction

There will be considerable growth in the aging population during the coming decades. Aging results in structural and functional impairments in ocular vascular networks [1], and

Ethics Committee for Tokyo Institute of Technology has imposed them. The contact info for the IRB of Tokyo Institute of Tech is hitorinri@jim.titech.ac.jp and may be contacted for data access requests.

**Funding:** This work was supported by JSPS KAKENHI Grant Number 21K11643 to NH. The funders had no role in study design, data collection and analysis, decision to publish, or preparation of the manuscript.

**Competing interests:** The authors have declared that no competing interests exist.

consequently increases the risk of vision impairment [2], especially among those older than 50 years [3]. In countries with aging societies such as Japan, eye diseases such as diabetic retinopathy, glaucoma, and age-related macular degeneration are relatively common. Preventing ocular vascular dysfunction can improve the quality of life.

Physical exercise habits may benefit people with visual diseases [4]. People who engage in physical activity had lower rates of glaucomatous vision loss [5]. Another correlational study analyzed over 8000 children's data in Ireland, revealing that refractive error and vision problems were significantly associated with increased sedentary behavior and decreased physical activity [6]. Nevertheless, effects of exercise habits on preventing ocular vascular dysfunction is still unclear.

Exercise habits have been shown to improve vascular stiffness not only in the limbs but also many other regions of the body [7], whereas effects of exercise habits on the ocular circulation remain unknown Effects of exercise habits on improving vascular stiffness can vary between individuals and body [8]. Exercise habits are strongly recommended, but a large proportion of adults cannot meet these recommendations, including in Japan [9]. This situation indicates the importance of research into effect of exercise habits on ocular circulation. Thus, the present cross-sectional study examined the effects of aging and exercise habits on the ocular blood flow (OBF) and its profiles (i.e., indices of ocular vascular flexibility) in the optic nerve head (ONH) and choroidal regions in healthy participants covering a wide age range. We hypothesized that the stiffness of the ocular circulation increases with aging and that exercise habits can decrease the stiffness, because they have been shown to be effective at many other sites [10].

## Methods

The present study was approved by the Research Ethics Committee for Tokyo Institute of Technology (approval number 2018036). All of the protocols used conformed with the standards set by the Declaration of Helsinki. Each participant received verbal and written explanations of the objectives, measurements, and risks and benefits associated with this study, after which written informed consent was obtained.

### Participants and study protocol

We used the data obtained from 1,079 participants in the medical checkup program at the Department of Health Care Center of the Japan Community Health Care Organization, Tokyo Kamata Medical Center. Participants who had metabolic syndrome as identified based on the Japanese committee to Evaluate Diagnostic Standards for Metabolic Syndrome were included [11, 12]. Participants were excluded if they had an ophthalmic disease, such as glaucoma, uveitis, optic neuropathy, vitreous or retinal disease, or retinal and choroidal vascular diseases; or atherosclerotic diseases such as hypertension, dyslipidemia, diabetes mellitus, cardiovascular or cerebrovascular events, arrhythmia, or exclusion criteria and a best-corrected visual acuity of <40/50; or if they had undergone a previous intraocular surgery. And we had also confirmed the medical history of the drug by inquiry and the eye disease by the photograph of the fundus.

The study criteria were finally met by 880 participants (698 males and 182 females), and the measurements were made from December 2016 to December 2018. And participants who had metabolic syndrome were met by 177 participants (172 males and 5 females).

The participants were categorized into three groups according to their age: young (<40 years), middle-aged (41–64 years), and elderly (>65 years). The participants were also categorized into two groups according to their exercise habits (Ex group and Non-ex group) as

determined based on two criteria: (1) self-reporting in a questionnaire and (2) the definition of the Ministry of Health, Labor and Welfare of Japan (MHLW). In the MHLW definition, exercise habits are present when exercising more than twice a week for more than 30 minutes at a time and for more than 1 year.

The participants were instructed to not consume caffeine-containing or alcohol-containing beverages or spicy food, and to not perform high-intensity exercise for at least 1 day prior to the measurements. The participants were also instructed to not smoke on the experimental day. All of the evaluations were performed between 0900 and 1100 hours, after the participants had fasted overnight. Participants were allowed to wear contact lens but not glasses during the OBF measurement, with pupil-dilating eye drops not being used. Excellent repeatability was previously demonstrated between the mean blur rate (MBR) measurements made with and without pupil dilation [13].

## Measurements

**Ocular circulation.**   The participants first rested for 10 minutes in a room at a temperature of 24˚C. The face was then fixed on a measurement base, and the blood flow profile in the right eye was measured for 4 seconds in the seated position using laser speckle flowgraphy (LSFG).

**Laser speckle flowgraphy.**   LSFG was applied using a system (LSFG-NAVI, Softcare, Fukuoka, Japan) developed to measure OBF and assess the OBF profile [14]. LSFG assesses Mean Blur Rate (MBR), which reflects the relative blood flow velocity and is correlated with the actual blood flow volume as measured using hydrogen gas clearance and microsphere methods [15].

We assessed the blood flow profile using LSFG Analyzer software (version 3.0.47, Softcare). The detail of the analysis was similar to those reported previously [16, 17]. The software program then separated out the vessels using an automated definitive threshold and divided the ONH and choroid into the vessel area and the capillary area. The following indices were calculated from the obtained data: mean blur rate (MBR, which reflects the OBF velocity), blowout time (BOT), and blowout score (BOS). These variables were analyzed separately in the ONH tissue (Tissue), in the ONH vessels (Vessel), throughout the ONH (All), and in the choroid (ChBFlow). BOS and BOT have previously been proposed as OBF profiles for evaluating vascular flexibility.

MBR is a measure of relative blood flow velocity and is expressed in arbitrary units (AU). It is calculated from the speckle pattern in the LSFG produced by the interference of laser light scattered by moving red blood cells in the ocular blood vessels.

BOT represents the proportion of time that the waveform is higher than half of the mean of the minimum and maximum signals during one heartbeat. A high BOT indicates that a high blood flow is maintained for a longer time during one heartbeat, indicating greater nutrition being supplied to the periphery.

BOS is another value developed to evaluate the amount of blood flowing through a blood vessel during one heartbeat. BOS is an index of the blood flow that is maintained between heartbeats and is calculated from the difference between the maximum and minimum MBRs as well as the average MBR. A high BOS indicates a high constancy of blood flow during the heartbeat, and is related to vascular resistance [18].

**Systemic hemodynamics and ophthalmic examinations.**   The systolic blood pressure (SBP), diastolic blood pressure (DBP), heart rate (HR), intraocular pressure (IOP), red blood cell (RBC) count, fasting blood glucose (FBG), and triglyceride (TG), high-density lipoprotein cholesterol (HDL), and low-density lipoprotein cholesterol (LDL) levels were measured and

the blood analysis performed at Department of Health Care Center of the Japan Community Health Care Organization, Tokyo Kamata Medical Center.

The participants were also asked about their smoking status. Blood pressure was recorded as the mean of two measurements made using a commercial sphygmomanometer after the participants had been seated for 10 minutes. Height and body mass were measured with participants wearing light clothing without shoes. Waist circumference was measured by using a flexible inch tape. The Body Mass Index (BMI) was calculated as body mass (kg) / height$^2$ (m).

IOP was measured using a commercial sphygmomanometer (NIDEK, Aichi, Japan). The mean arterial pressure (MAP) was calculated as DBP + 1/3(SBP–DBP). The mean ocular perfusion pressure (OPP) was calculated as 2/3(MAP–IOP) [19]. This formula is based on evidence that the pressure in choroidal veins almost equals IOP [20].

**Questionnaire on exercise habits.** A questionnaire was used to estimate the presence of exercise habits, based on the frequency of exercise per week, duration of each exercise session, exercise intensity, exercise history, step per day and exercise types (Supporting information).

## Statistical analysis

Statistical analysis was performed using standard statistical software (version 25.0, SPSS Statistics, IBM Corporation, Armonk, NY, USA). Data are expressed as mean ± SD values. In all statistical analyses the cutoff for significance was 5%. ANOVA, the Kruskal-Wallis test, and the Mann-Whitney U test were used to compare normally distributed data. Spearman's rank test was used to determine the coefficients for the correlations between the variables. Multiple logistic regression analysis was used to determine the associations between sex and the parameters of the pulse waveform analyses.

To examine the effects of aging and exercise habits, two-way ANOVA was used to evaluate the interactions of the OBF profile among groups. If a significant interaction is obtained, the two factors are considered to influence each other and hence cannot be separated. Bonferroni analysis was used as a post-hoc test to examine simple main effects.

Multiple linear regression analysis was used to identify which of the following factors independently affected the OBF profiles: sex, BMI, exercise habits, age squared, current smoking habit, IOP, HR, and the interaction of exercise habits and age squared.

## Results

The characteristics in the three age groups of this study are presented in Table 1. The height, body mass, BMI, waist circumference, RBC count, FBG, TG, LDL, and current smoking rate differed significantly among the three age groups. SBP and MAP were significantly higher in the elderly group, whereas DBP was significantly higher in the middle-aged group.

MBR-All, MBR-Vessel, and all sections of the BOS and BOT were significantly lower in the elderly groups. OPP was significantly higher in the elderly group. The number of steps per day was significant larger in the middle-aged group, while the average exercise intensity decreased significantly as age increased.

The characteristics in the two groups of self-reported exercise habits are presented in Table 2. Those in the Ex group were significantly older (51.0±9.5 years) than those in the Non-ex group (49.2±8.9 years). The current smoking rate was significantly higher in the Non-ex group than in the Ex group. The HDL level was significantly higher while the TG level was significantly lower in the Ex group than in the Non-ex group.

MBR-All was significantly higher in the Non-ex group than in Ex group. All sections of the average BOS and BOT. IOP and OPP did not differ significantly between the two groups. The

**Table 1. Characteristics in the three age groups.**

| | Young (n = 140) | Middle-aged (n = 676) | Elderly (n = 64) | P |
|---|---|---|---|---|
| Age, years | 37.5±2.7 | 50.6±6.3 | 69.5±3.7 | <0.01*†‡ |
| Height, cm | 168.5±7.7 | 168.8±7.3 | 163.3±7.8 | <0.01†‡ |
| Body mass, kg | 66.3±13.9 | 68.6±12.0 | 62.5±8.3 | <0.01*‡ |
| BMI, kg/m$^2$ | 23.2±4.1 | 24.0±3.4 | 23.4±2.5 | <0.01* |
| Waist circumference, cm | 81.2±11.3 | 84.6±9.6 | 83.9±7.5 | <0.01*† |
| SBP, mmHg | 116.7±15.8 | 124.3±18.0 | 134.5±15.4 | <0.01*†‡ |
| DBP, mmHg | 71.8±11.6 | 78.2±13.2 | 77.3±11.2 | <0.01*† |
| MAP, mmHg | 86.8±12.5 | 93.5±14.1 | 96.4±11.6 | <0.01*† |
| HR, bpm | 70.9±10.6 | 70.7±10.0 | 71.1±11.0 | 0.94 |
| RBC, × 10$^4$/μl | 4.8±0.5 | 4.8±0.5 | 4.6±0.5 | <0.01†‡ |
| FBG, mg/dL | 95.4±16.5 | 102.0±15.8 | 110.7±20.0 | <0.01*†‡ |
| TG, mg/dL | 108.9±78.2 | 131.5±99.8 | 114.2±61.7 | 0.02* |
| HDL, mg/dL | 63.3±17.6 | 63.6±17.1 | 64.8±16.7 | 0.75 |
| LDL, mg/dL | 120.0±30.5 | 131.6±31.5 | 133.1±26.9 | <0.01*† |
| IOP, mmHg | 11.9±2.4 | 12.0±2.7 | 11.9±2.8 | 0.96 |
| OPP, mmHg | 46.0±8.1 | 50.4±9.5 | 52.3±8.1 | <0.01*† |
| MBR-All, au | 26.1±4.0 | 25.4±4.5 | 23.4±5.0 | <0.01†‡ |
| BOS-All | 81.4±3.6 | 80.7±4.1 | 73.9±5.1 | <0.01†‡ |
| BOT-All | 54.9±3.8 | 52.2±3.7 | 47.9±3.8 | 0.01*†‡ |
| MBR-Tissue, au | 13.0±2.4 | 13.0±2.5 | 12.8±3.1 | 0.83 |
| BOS-Tissue | 78.9±3.7 | 77.8±4.4 | 70.3±5.8 | <0.01*†‡ |
| BOT-Tissue | 52.2±4.2 | 49.3±3.8 | 44.9±3.7 | <0.01*†‡ |
| MBR-Vessel, au | 46.0±7.3 | 45.4±7.0 | 42.6±7.4 | <0.01†‡ |
| BOS-Vessel | 82.4±3.7 | 81.9±4.1 | 75.8±4.8 | <0.01†‡ |
| BOT-Vessel | 56.0±3.8 | 53.8±3.9 | 49.8±3.8 | <0.01*†‡ |
| MBR-ChBFlow, au | 9.7±3.1 | 9.2±3.0 | 9.5±3.5 | 0.19 |
| BOS-ChBFlow | 78.6±4.0 | 76.9±5.0 | 69.8±6.0 | <0.01*†‡ |
| BOT-ChBFlow | 51.3±4.2 | 48.4±3.9 | 44.5±3.2 | <0.01*†‡ |
| Step per day | 5663±3005 | 6697±3667 | 5700±3491 | 0.02*‡ |
| Exercise intensity | 5.9±2.2 | 5.4±2.2 | 4.4±1.9 | 0.02† |
| Current smoking (%) | 39 (27.9) | 219 (32.4) | 9 (14.1) | 0.01*†‡ |

Data are presented as mean ± SE.

P-value was come from one-way analysis of variance (one-way ANOVA) and Kruskal-Wallis test.

* Indicates statistically significant difference between the young and the middle-aged;

† indicates statistically significant difference between the young and the elderly;

‡ indicates statistically significant difference between the middle-aged and the elderly (Dunnet test).

number of steps per day was significantly larger in the Ex group than in the Non-ex group. The exercise intensity in the Ex group was 5.4±2.2 (out of 10).

The characteristics in the two groups categorized based on the MHLW definition (n = 192 and 688 in the Ex and Non-ex groups, respectively) in this study are presented in Table 3. Those in the Ex group were significant older (52.0±9.8 years) than those in the Non-ex group (49.3±8.9 years). The current smoking rate was significantly higher in the Non-ex group than in the Ex group. The HDL level was significantly higher while the TG level was significantly lower in the Ex group than in the Non-ex group. The HR was significantly higher in the Non-ex group than in the Ex group.

**Table 2. Characteristics categorized by the self-reported Ex and Non-ex groups.**

| | Ex group (n = 321) | Non-ex group (n = 559) | P |
|---|---|---|---|
| Age, years | 51.0±9.5 | 49.2±9.0 | <0.01 |
| Height, cm | 168.0±7.7 | 168.5±7.4 | 0.31 |
| Body mass, kg | 68.0±11.5 | 67.6±12.6 | 0.97 |
| BMI, kg/m2 | 24.0±3.3 | 23.7±3.6 | 0.31 |
| Waist circumference, cm | 80.0±8.7 | 84.1±9.9 | 0.34 |
| SBP, mmHg | 124.8±16.8 | 123.3±17.5 | 0.18 |
| DBP, mmHg | 76.8±12.8 | 77.3±13.1 | 0.63 |
| MAP, mmHg | 92.8±14.7 | 92.6±13.9 | 0.82 |
| HR, bpm | 69.2±10.1 | 71.7±10.0 | <0.01 |
| RBC, × $10^4$/μl | 4.8±0.4 | 4.8±0.5 | 0.47 |
| FBG, mg/dL | 101.8±15.6 | 101.5±17.2 | 0.78 |
| TG, mg/dL | 115.2±69.1 | 133.2±106.1 | 0.01 |
| HDL, mg/dL | 66.5±17.8 | 62.0±16.5 | <0.01 |
| LDL, mg/dL | 127.9±29.1 | 131.0±32.5 | 0.22 |
| IOP, mmHg | 11.9±2.6 | 11.9±2.7 | 0.33 |
| OPP, mmHg | 50.0±9.8 | 49.8±9.2 | 0.78 |
| MBR-All, au | 24.9±4.5 | 25.6±4.5 | 0.04 |
| BOS-All | 79.8±4.5 | 80.6±4.5 | 0.03 |
| BOT-All | 51.9±4.0 | 52.6±4.1 | 0.01 |
| MBR-Tissue, au | 12.8±2.4 | 13.0±2.6 | 0.31 |
| BOS-Tissue | 76.8±5.0 | 77.8±4.8 | 0.03 |
| BOT-Tissue | 49.0±4.2 | 49.7±4.2 | 0.01 |
| MBR-Vessel, au | 44.9±7.1 | 45.5±7.2 | 0.24 |
| BOS-Vessel | 81.2±4.4 | 81.8±4.4 | 0.01 |
| BOT-Vessel | 53.4±4.1 | 54.1±4.1 | 0.01 |
| MBR-ChBFlow, au | 9.2±3.2 | 9.4±3.0 | 0.08 |
| BOS-ChBFlow | 76.0±5.4 | 77.0±5.3 | 0.02 |
| BOT-ChBFlow | 48.1±4.1 | 48.8±4.3 | <0.01 |
| Step per day | 7359±3883 | 5975±3318 | <0.01 |
| Exercise intensity | 5.4±2.2 | | |
| Current smoking (%) | 72 (22.4) | 195(34.9) | 0.03 |

Data are presented as mean ± SE.

P-value was come from two-sample t-test and Mann Whitney U test.

MBR-All, MBR-Vessel, BOS-All, BOS-Vessel, BOS-ChBFlow, and BOT in all sections were significantly higher in the Non-ex group than in the Ex group, while BOS-Tissue was significantly lower in the Non-ex group. The number of steps per day was significantly larger in the Ex group than in the Non-ex group. The exercise intensity in the Ex group was 5.6±2.2.

The characteristics of the 698 males and 182 females in this study are shown in Table 4. The age of the male (50.1 ± 9.1 yrs) did not differ significantly from those of the female (49.1 ± 9.6 yrs). The height, body mass, BMI, waist circumference, RBC count, FBG, TG and current smoking rate of the male were all significantly higher than those of the female. The HDL of the female was significantly higher than those of the male, whereas the LDL and TG in the male were significantly higher than those in the female. SBP, DBP and MAP of the male were significantly higher than those of the female.

**Table 3. Characteristics categorized by the MHLW-based Ex and Non-ex groups.**

| | Ex group (n = 192) | Non-ex group (n = 688) | P |
|---|---|---|---|
| Age, years | 52.0±9.8 | 49.3±8.9 | 0.01 |
| Height, cm | 167.6±7.8 | 168.5±7.5 | 0.17 |
| Body mass, kg | 67.8±11.7 | 67.8±12.4 | 0.59 |
| BMI, kg/m$^2$ | 24.1±3.4 | 23.8±3.5 | 0.16 |
| Waist circumference, cm | 83.6±8.8 | 84.2±9.7 | 0.86 |
| SBP, mmHg | 125.3±17.9 | 123.5±17.9 | 0.3 |
| DBP, mmHg | 77.2±12.1 | 77.1±13.2 | 0.96 |
| MAP, mmHg | 93.2±13.2 | 92.6±14.1 | 0.58 |
| HR, bpm | 68.1±9.8 | 71.5±10.1 | <0.01 |
| RBC, × 10$^4$/µl | 4.8±0.5 | 4.8±0.5 | 0.42 |
| FBG, mg/dL | 101.8±15.2 | 101.5±17.0 | 0.85 |
| TG, mg/dL | 109.8±64.3 | 131.7±101.6 | 0.04 |
| HDL, mg/dL | 69.5±19.3 | 62.0±16.2 | <0.01 |
| LDL, mg/dL | 128.3±30.3 | 130.4±31.7 | 0.08 |
| IOP, mmHg | 12.1±2.6 | 11.9±2.7 | 0.64 |
| OPP, mmHg | 50.2±9.1 | 49.8±9.4 | 0.64 |
| MBR-All, au | 24.5±4.5 | 25.6±4.5 | <0.01 |
| BOS-All | 79.6±4.6 | 80.5±4.4 | 0.01 |
| BOT-All | 51.4±4.3 | 52.6±4.0 | <0.01 |
| MBR-Tissue, au | 12.7±2.5 | 13.0±2.5 | 0.09 |
| BOS-Tissue | 78.1±4.7 | 74.9±4.5 | 0.01 |
| BOT-Tissue | 48.6±4.4 | 49.7±4.1 | <0.01 |
| MBR-Vessel, au | 43.8±6.6 | 45.7±7.2 | <0.01 |
| BOS-Vessel | 81.0±4.3 | 81.7±4.4 | 0.04 |
| BOT-Vessel | 53.0±4.4 | 54.1±4.0 | <0.01 |
| MBR-ChBFlow, au | 9.0±3.1 | 9.4±3.0 | 0.08 |
| BOS-ChBFlow | 75.7±5.6 | 76.9±5.3 | <0.01 |
| BOT-ChBFlow | 47.7±4.4 | 48.7±4.2 | <0.01 |
| Step per day | 8069±4199 | 6013±3287 | <0.01 |
| Exercise intensity | 5.6±2.2 | | |
| Current smoking (%) | 32 (16.7) | 232 (34.2) | <0.01 |

Data are presented as mean ± SE.

P-value was come from two-sample t-test and Mann Whitney U test.

The MBR-All, MBR-Tissue and MBR-Vessel in the male were significantly lower than those in the female. All sections of the BOS and BOT in the male were significantly higher than those of the female. The exercise intensity of the male was significantly higher than that of the female.

There was no significant interaction of age and exercise habits on IOP, OPP, MBR-All, MBR-Tissue, MBR-Choroid, or BOT and BOS in all sections. There was a significant interaction of age and exercise habits on MBR-Vessel, significantly decreasing in the Non-ex group among all three age groups (Fig 1).

There was a significant interaction of age and exercise habits on BOT-All, which decreased significantly in the Ex group among all three age groups (Fig 2). There was no significant interaction of age and MHLW-based exercise habits on IOP, OPP, BOT-Vessel, or MBR and BOS

**Table 4. The characteristics of male and female participants.**

| | Male (n = 698) | Female (n = 182) | P |
|---|---|---|---|
| Age, years | 50.1±9.1 | 49.1±9.6 | 0.21 |
| Height, cm | 170.8±6.0 | 158.9±5.0 | <0.01 |
| Body mass, kg | 70.7±11.0 | 56.6±10.0 | <0.01 |
| BMI, kg/m$^2$ | 24.2±3.3 | 22.4±3.6 | <0.01 |
| Waist circumference, cm | 85.4±8.9 | 78.8±9.8 | <0.01 |
| SBP, mmHg | 125.3±17.8 | 118.3±17.4 | <0.01 |
| DBP, mmHg | 78.9±12.6 | 70.0±12.0 | <0.01 |
| MAP, mmHg | 94.4±13.7 | 86.1±12.8 | <0.01 |
| HR, bpm | 70.9±10.3 | 70.5±9.4 | 0.71 |
| RBC, × 10$^4$/µl | 4.9±0.4 | 4.4±0.4 | <0.01 |
| FBG, mg/dL | 103.4±17.5 | 94.6±9.8 | <0.01 |
| TG, mg/dL | 136.2±101.6 | 90.0±45.8 | <0.01 |
| HDL, mg/dL | 60.5±15.3 | 75.8±18.2 | <0.01 |
| LDL, mg/dL | 131.7±30.4 | 122.9±33.9 | <0.01 |
| IOP, mmHg | 12.0±2.7 | 11.7±2.6 | 0.11 |
| OPP, mmHg | 50.9±9.2 | 45.7±8.7 | <0.01 |
| MBR-All, au | 24.8±4.5 | 27.4±3.9 | <0.01 |
| BOS-All | 80.9±4.3 | 77.9±4.4 | <0.01 |
| BOT-All | 52.6±4.0 | 51.5±4.1 | 0.01 |
| MBR-Tissue, au | 12.8±2.6 | 13.6±2.3 | <0.01 |
| BOS-Tissue | 78.1±4.7 | 74.9±4.5 | <0.01 |
| BOT-Tissue | 49.7±4.2 | 48.5±4.0 | <0.01 |
| MBR-Vessel, au | 44.6±7.0 | 48.2±6.9 | <0.01 |
| BOS-Vessel | 82.1±4.2 | 79.1±4.3 | <0.01 |
| BOT-Vessel | 54.1±4.1 | 53.1±4.0 | <0.01 |
| MBR-ChBFlow, au | 9.4±3.1 | 9.2±3.1 | 0.62 |
| BOS-ChBFlow | 77.2±5.4 | 74.7±4.8 | <0.01 |
| BOT-ChBFlow | 48.7±4.3 | 47.9±3.9 | 0.01 |
| Step per day | 6567±3718 | 6079±2977 | 0.01 |
| Exercise intensity | 5.5±2.2 | 4.8±2.0 | 0.42 |
| Current smoking (%) | 238(34.1) | 29(15.9) | 0.02 |

Data are presented as mean ± SE.

P-value was come from two-sample t-test and Mann Whitney U test.

in all sections. BOT-All in the Non-ex group was significantly higher in the middle-aged group.

There tended to be significant interaction of age and exercise habits on BOS-Tissue. BOS-Tissue decreased significantly with age in both the Ex and Non-ex groups (p = 0.055, Fig 3).

Table 5 presents the standardized-coefficient and R-squared values for the systemic hemo-dynamics, ophthalmic observations, and exercise habits obtained in a multiple linear regression analysis. We determined the best regression formula for fitting the data on sex, BMI, self-reported exercise habits, age squared, current smoking, IOP, HR, exercise habits × age squared, and steps per day.

Table 6 presents the standardized-coefficient and R-squared values for systemic hemody-namics, ophthalmic observations, and exercise habits obtained in a multiple linear regression analysis. We determined the best regression formula for fitting the data on sex, BMI, MHLW-

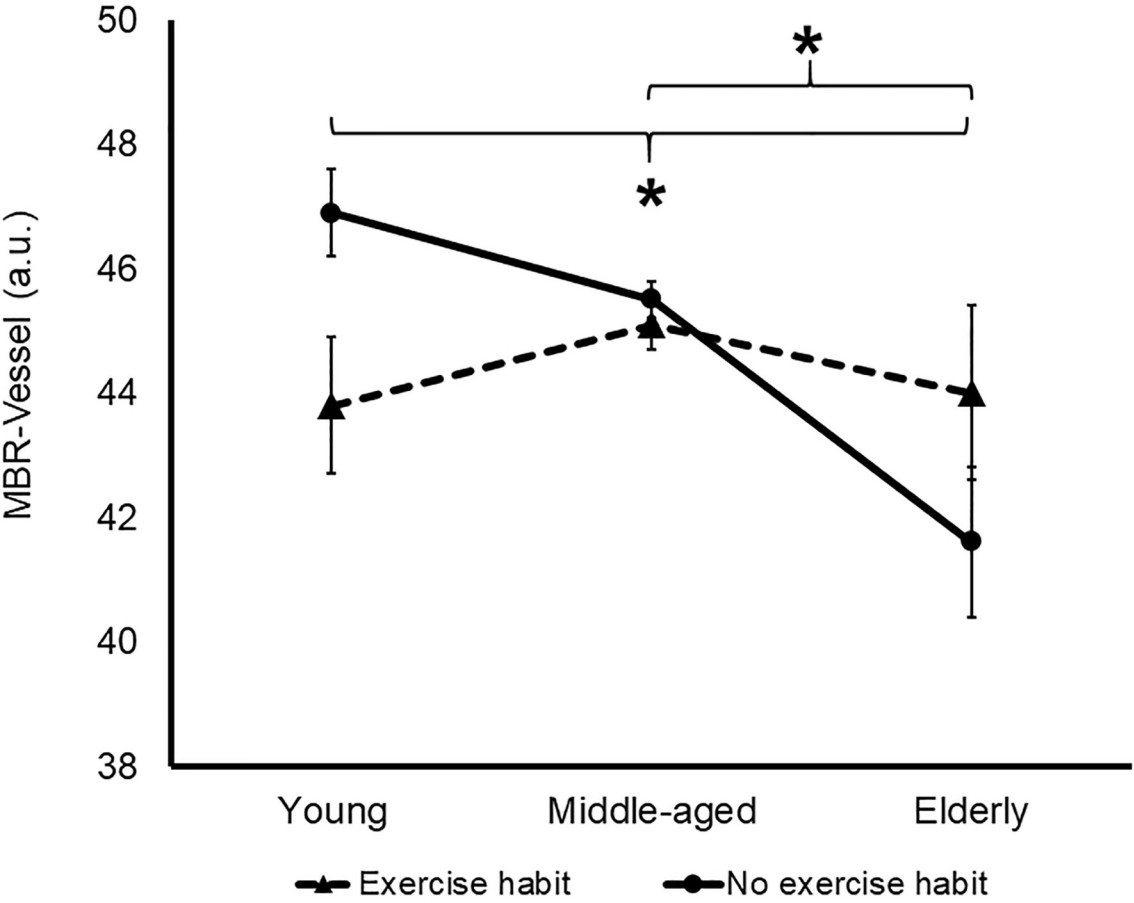

**Fig 1. Effects of aging and self-reported exercise habits on MBR-Vessel.** *, significant difference among young (<40 years old), middle-aged (41–64) and elderly (>65) groups without exercise habits; †, significant difference between groups without exercise habits. Data are mean and SD.

based exercise habits, age squared, current smoking, IOP, exercise habits × age squared, and steps per day.

## Discussion

We have investigated the effects of aging and exercise habits on blood flow profiles throughout the ONH and choroidal arteries in a large population of participants in yearly medical check-ups. The main findings of the present cross-sectional study were as follows: (1) an effect of aging on ocular circulation was demonstrated in a large population, supporting previous studies, (2) beneficial effects of exercise habits on ocular circulation were not supported; in contrast to our hypothesis, the ocular flow and profiles showed trends of vessel stiffening in the Ex group, and (3) ocular vessels were stiffer in males than in females.

BOT and BOS in various areas of the ocular circulation decrease with aging, regardless of the type of ocular vasculature. BOT and BOS throughout the ONH (i.e., All, Vessel, and Tissue) and the choroidal artery decreased with age, in accordance to several reports of vascular

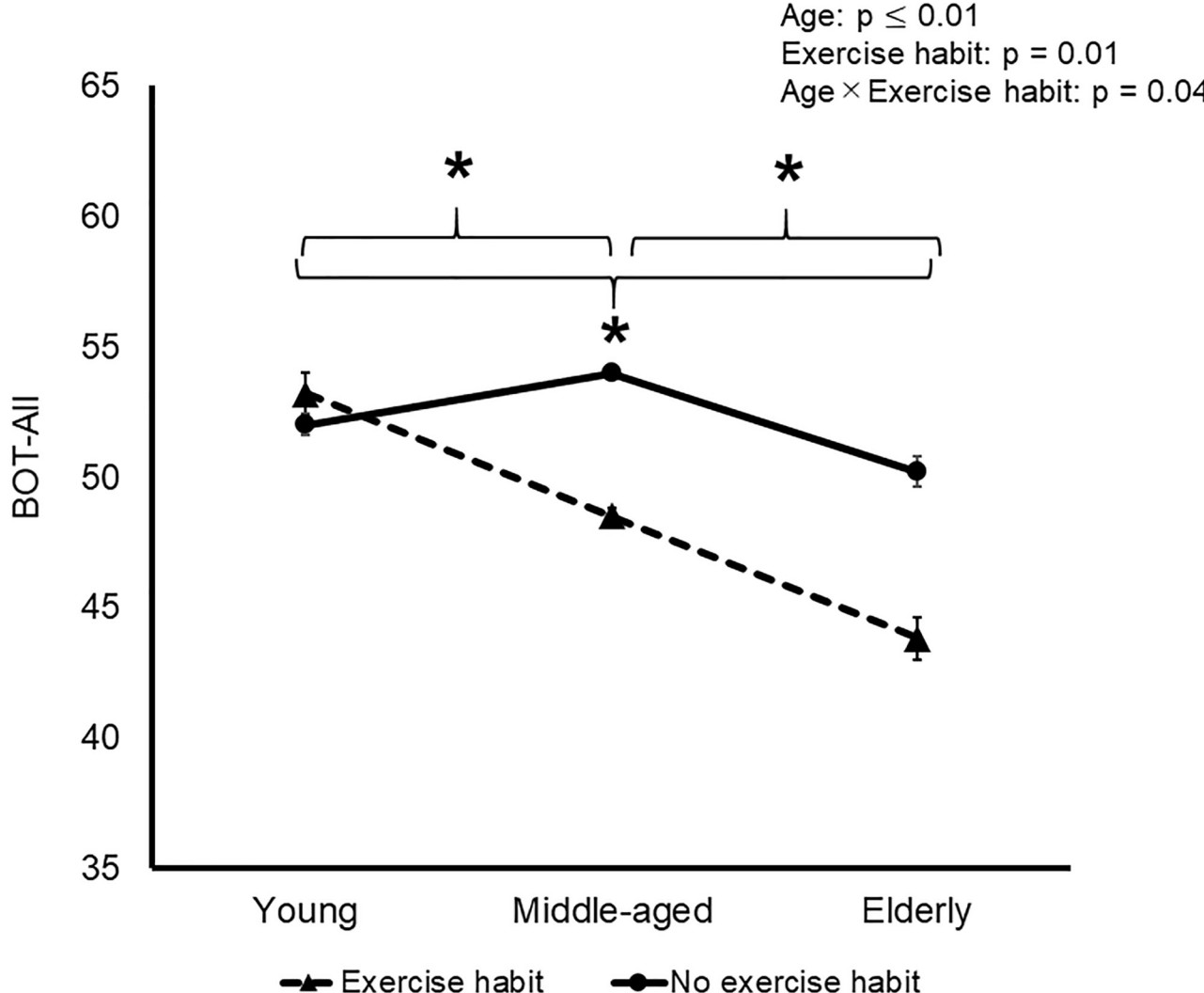

**Fig 2. Effects of aging and MHLW-based exercise habits on BOT-All.** *, significant difference among young (<40 years old), middle-aged (41–64) and elderly (>65) groups with and without exercise habits. age group. Data are mean and SD.

vessels throughout the ONH stiffening with age [18, 21, 22]. These results confirm the effects of age on vessel stiffness in the ocular circulation.

The present study found no preferable OBF profiles in the Ex group compared with the Non-ex group when exercise habits were assessed either in a self-reported manner or according to the MHLW definition. In short, exercise habits do not appear to improve the ocular circulation. A lack of exercise habits greatly influences the risk of vessel stiffening [23], whereas exercise habits improve the stiffness not only in limbs but also in many other regions of the body [4]. There are various mechanisms underlying how exercise habits can reduce the risks of stiffening and cerebrovascular diseases [24]. The TG and HDL levels, which are risk factors that exercise habits can improve, were lower in the Ex group. Nevertheless, exercise habits were not correlated with improved a blood flow profile in ocular vessels.

The effects of exercise habits on human arterial stiffness vary according to the site and size of the arteries [25]. Many previous studies have showed that exercise habits can reduce central

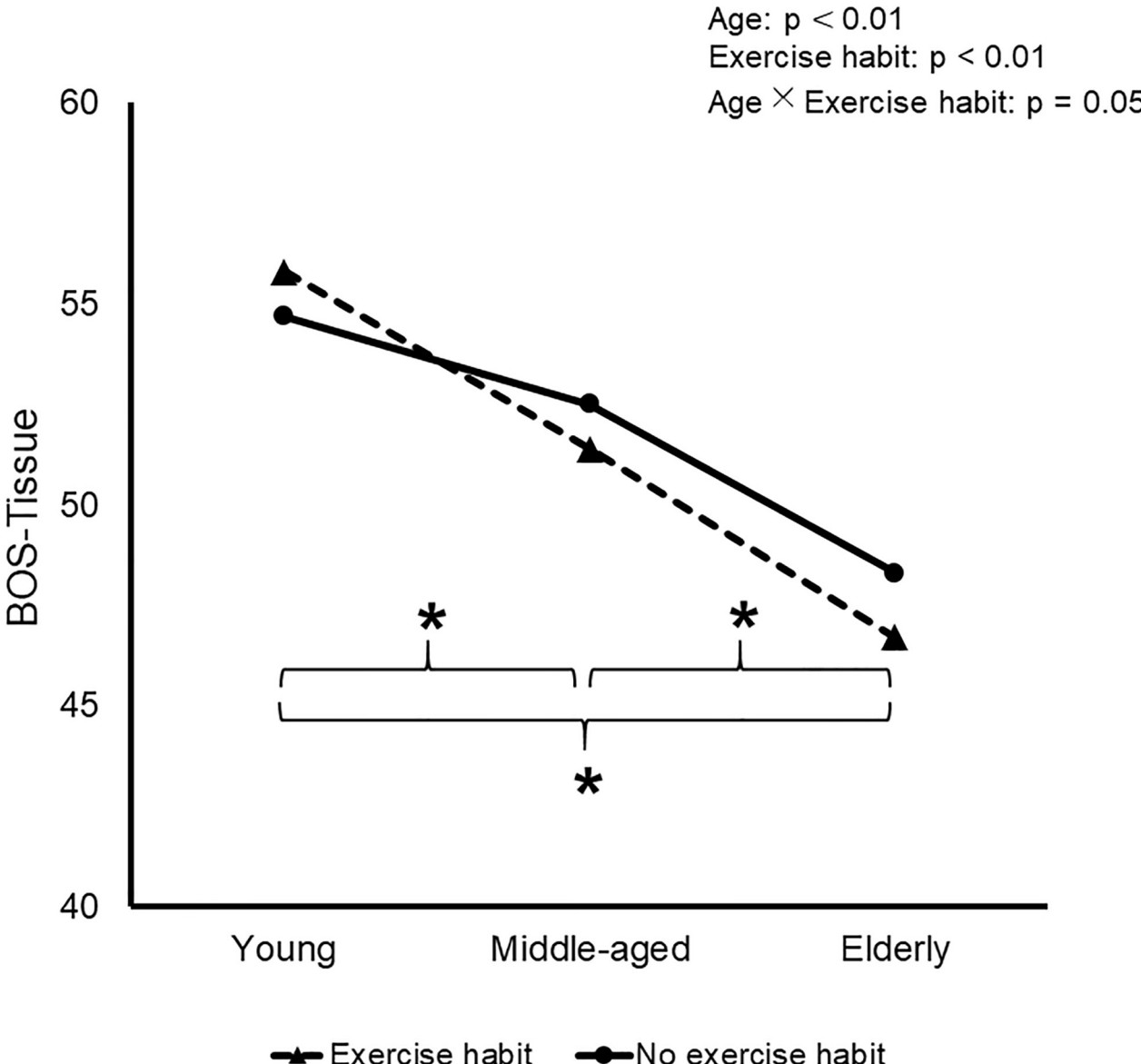

Age: p < 0.01
Exercise habit: p < 0.01
Age × Exercise habit: p = 0.055

**Fig 3. Effects of aging and MHLW-based exercise habits on BOS-Tissue.** *, significant difference among young (<40 years old), middle-aged (41–64) and elderly (>65) groups with and without exercise habits. Data are mean and SD.

arterial stiffness [26], especially that of large central arteries [25]. Exercising four or five times weekly is associated with reduced central arterial stiffness in older people. Vascular stiffness appears to be lowest in elderly who perform habitual physical activity of at least around 6,600 steps/day and/or spend more than 16 minutes/day performing exercise at an intensity of >3 metabolic equivalents [27]. Casual exercise training of two or three times weekly may be sufficient to minimize the stiffening of middle-sized arteries such as the carotid that is associated with aging [25]. However, exercise habits have only minor effects on small peripheral arteries [25], age-related decreases in basal limb blood flow and vascular conductivity [28], and chronically elevated arterial pressure [29, 30]. In contrast to large and central arteries and middle-sized arteries, the present study found a rather nonpreferable effect of exercise on BOT, as

**Table 5. Results of the multiple regression analysis for factors independently contributing to the item of ocular microcirculation, using the self-reported exercise habits.**

| | $R^2$ | Constant | Exercise habit | Steps per day | Sex(M:0,F;1) | $Age^2$ | BMI | Current smoking | IOP | Exercise habit*$Age^2$ | HR |
|---|---|---|---|---|---|---|---|---|---|---|---|
| **MBR-Vessel** | 0.044 | 45.338 | -2.001* | 0.329 | -0.864** | -1.521*** | -0.317 | 0.453 | 0.010 | 1.759† | -0.032 |
| **MBR-Tissue** | 0.027 | 12.964 | -0.026 | 0.017 | -0.159 | -0.015 | -0.404*** | 0.101 | 0.311** | -0.012 | -0.036 |
| **MBR-All** | 0.068 | 25.367 | -0.914 | 0.017 | -0.760*** | -0.826*** | -0.540** | 0.228 | 0.125 | 0.771 | 0.088 |
| **BOS-Vessel** | 0.478 | 81.578 | 0.628 | 0.113 | 1.309*** | -1.579*** | 0.306* | 0.035 | -0.507*** | -0.690 | 2.039*** |
| **BOS-Tissue** | 0.559 | 77.551 | 0.958* | 0.140 | 1.458*** | -2.075*** | 0.178 | -0.134 | -0.388† | -1.215** | 2.366*** |
| **BOS-All** | 0.529 | 80.360 | 0.708† | 0.132 | 1.325*** | -1.795*** | 0.271† | -0.015 | -0.489*** | -0.837* | 2.165*** |
| **BOT-Vessel** | 0.330 | 53.860 | 0.070 | -0.053 | 0.488** | -1.669*** | 0.034 | 0.156 | -0.293* | -0.266 | 1.284*** |
| **BOT-Tissue** | 0.460 | 49.552 | 0.466 | -0.107 | 0.672*** | -2.000*** | -0.229 | -0.048 | -0.097 | -0.488 | 1.692*** |
| **BOT-All** | 0.429 | 52.414 | 0.115 | -0.038 | 0.532*** | -1.894*** | -0.039 | 0.051 | -0.200 | -0.255 | 1.489*** |
| **MBR-ChBFflow** | 0.023 | 9.346 | -1.152** | 0.155 | 0.073 | -0.302† | -0.417** | -0.107 | 0.141 | 0.982* | 0.058 |
| **BOS-ChBFflow** | 0.460 | 76.813 | 0.528 | 0.180 | 1.162*** | -2.263*** | -0.077 | 0.013 | -0.475** | -0.674 | 2.556*** |
| **BOT-ChBFflow** | 0.419 | 48.555 | 0.526 | 0.125 | 0.548*** | -1.717*** | -0.226 | -0.042 | -0.299* | -0.662 | 1.739*** |

*: $p<0.05$;

**: $p<0.01$;

***: $p<0.001$;

†: $0.05 < p < 0.08$

shown by an interaction in BOT-All and a marginally significant interaction (p = 0.055) in BOS-Tissue of age and MHLW-based exercise habits. BOS-Tissue and BOT-All of the elderly were lower in the Ex group than in the Non-ex group. The results suggest that the stiffness of ocular vessels cannot be improved by exercise habits.

**Table 6. Results of the multiple regression analysis for factors independently contributing to the item of ocular microcirculation, using the MHLW-based exercise habit.**

| | $R^2$ | Constant | Exercise habit | Steps per day | Sex(M:0,F;1) | $Age^2$ | BMI | Current smoking | IOP | Exercise habit*$Age^2$ | HR |
|---|---|---|---|---|---|---|---|---|---|---|---|
| **MBR-Vessel** | 0.050 | 45.342 | -2.382* | 0.409 | -0.797* | -1.418*** | -0.326 | 0.282 | -0.031 | 1.763† | 0.013 |
| **MBR-Tissue** | 0.026 | 12.976 | -0.153 | 0.018 | -0.153 | -0.051 | -0.408*** | 0.088 | 0.295** | 0.114 | -0.025 |
| **MBR-All** | 0.068 | 25.386 | -0.881 | 0.023 | -0.745*** | -0.797*** | -0.560** | 0.181 | 0.095 | 0.735 | 0.129 |
| **BOS-Vessel** | 0.479 | 81.616 | 0.798† | 0.124 | 1.300*** | -1.590*** | 0.305* | 0.064 | -0.535*** | -0.780† | 2.052*** |
| **BOS-Tissue** | 0.558 | 77.596 | 1.364** | 0.123 | 1.439*** | -2.128*** | 0.169 | -0.073 | -0.409** | -1.378** | 2.377*** |
| **BOS-All** | 0.529 | 80.401 | 0.997* | 0.131 | 1.311*** | -1.814*** | †0.266† | 0.026 | -0.515*** | -0.979* | 2.179*** |
| **BOT-Vessel** | 0.331 | 53.842 | 0.008 | -0.017 | 0.471** | -1.650*** | 0.057 | 0.154 | -0.280† | -0.362 | 1.247*** |
| **BOT-Tissue** | 0.453 | 49.525 | 0.214 | -0.089 | 0.665*** | -2.041*** | -0.211 | -0.039 | -0.071 | -0.327 | 1.653*** |
| **BOT-All** | 0.427 | 52.393 | -0.058 | -0.006 | 0.519*** | -1.893*** | -0.018 | 0.047 | -0.185 | -0.223 | 1.456*** |
| **MBR-ChBFflow** | 0.012 | 9.365 | -0.675 | 0.116 | 0.058 | -0.228 | -0.450** | -0.102 | 0.125 | 0.687 | 0.103 |
| **BOS-ChBFflow** | 0.456 | 76.846 | 0.504 | 0.171 | 1.158*** | -2.336*** | -0.076 | 0.037 | -0.503** | -0.487 | 2.570*** |
| **BOT-ChBFflow** | 0.412 | 48.537 | 0.179 | 0.130 | 0.547*** | -1.814*** | -0.211 | -0.039 | -0.281* | -0.341 | 1.708*** |

*: $p<0.05$;

**: $p<0.01$;

***: $p<0.001$;

†: $0.05 < p < 0.08$

The self-reported exercise intensity was roughly 5 (out of a maximum of 10), which can be considered moderate. Moderate levels of physical activity are associated with lower risks of vascular diseases in systemic regions compared to inactivity [31]. To our best knowledge, there is no explanation for the apparent increase in risk for moderate exercise associated with the stiffness of the ocular circulation. Moderate exercise habits cannot fully explain the lack of effect of exercise habits on the ocular circulation.

Exercise habits might have a marginal effect on ocular circulation. An interaction of age and self-reported exercise habits on ocular vessels was found in this study solely for MBR-Vessel. MBR-Vessel was the highest in middle-aged people with exercise habits. MBR, which is automatically calculated from variations in the degree of blurring, is a quantitative index of the blood flow, and measurements of MBR are highly reproducible [32–34]. MAP is well known to increase with age, which is consistent with the present data (Table 1). MBR increases when MAP has increased. It is therefore expected that MAP would increase with aging, with MBR consequently also increasing. MBR-Vessel did not differ between the Ex and Non-ex groups. It was inferred that there remains a possibility that MBR was counterbalanced against the increase in MAP with age, possibly via a decrease in the stiffness of the ocular vessels.

It was particularly interesting that MBR-Vessel and BOT-All peaked in the middle-aged participants. A further longitudinal study is necessary to identify the factors that restrict the effects of exercise habits on improving stiffness with age throughout the ONH.

MBR decreased during an acute increase in IOP in the ONH tissue [35]. Changes in IOP were found to be related to changes in BOS [36]. Those previous studies indicated that IOP is consistent with BOS and MBR in the ocular circulation, supporting our results.

BOS-Vessel and BOS-All in the ONH and choroid were correlated with BMI. BMI increases not only the risk of cardiovascular disease in people who are metabolically unhealthy [37], but also the risk of stiffening in the ocular circulation.

IOP did not differ with age, which is inconsistent with a negative correlation reported between IOP and age among Japanese healthy participants [38]. Studies in Europe and the US found a positive correlation between IOP and age [39, 40]. The different population compositions of these studies could explain these discrepant results. The present study included both healthy participants and those with metabolic syndrome. Some previous studies have suggested that five components—waist circumference, high TG, high blood pressure, high FBG, and low HDL—in the presence of metabolic syndrome each showed a positive correlation with IOP [41]; in short, these components can influence IOP.

## Limitations

To the best of our knowledge, this is the first study to have investigated the effects of exercise habits on OBF over a wide range of ages in a large population. Nevertheless, our study was subject to some limitations that need to be considered when interpreting the findings. Firstly, the study population comprised more males than females, sex may influence the ocular circulation [16] which has been reported. Sex imbalance might have affected the results in our study. We did not survey the use of oral contraceptives or hormone replacement therapy. These effects should not be ignored. Also, some of the participants were smokers. Smoking habits may influence the ocular circulation [42], though we included smoking habits and history of smoking in the analysis parameters. Similarly, the effects of alcohol consumption was not studied. Secondly, we did not determine the detailed exercise habits of the participants, and so the present findings need to be confirmed in a longitudinal study. The exercise habits were estimated by the questionnaire (see Supporting information) where participants recorded the exercise habits by themselves. The recording apparatus for step count was not controlled. Thirdly, we did

not control for self-selection bias. Participants in a medical checkup program will be more health conscious than nonparticipants. Factors that probably affect the outcome of exercise habits were not assessed in the current study, such as dietary intake, the level of physical exercise in the Non-exercise group, social background, education level, and economic status. Fourthly, participants both in Ex and Non-ex groups had high fasting blood glucose (FBG). Blood flow in ophthalmic artery is decreased in patients with diabetes [43]. Fifthly, corneal thickness hasn't been measured in this study. Corneal thickness influences IOP. As thicker corneas lead to higher IOP and thinner lead to lower of IOP [44].

## Conclusion

The present study investigated the effects of age and exercise habits on the ocular circulation by assessing BOS and BOT (which are indices of vascular function) as well as MBR in a large population. Effects of age on the ocular circulation were found, demonstrating the stiffening of ocular vessels with age. However, exercise habits did not alter the ocular circulation, while there were significant interactions of age and exercise habits on MBR-Vessel and BOT-All, and a marginal significant interaction on BOS-Tissue. These results suggest that the stiffness of ocular vessels increases with aging and that exercise habits cannot improve them. Therefore, the hypothesis that exercise habits can improve the deterioration in the stiffness of the ocular circulation with aging is rejected.

## Supporting information

**S1 Questionnaire. Questionnaire on exercise habits.**
(DOCX)

## Acknowledgments

Support was received for this study from the Department of Health Care Center of the Japan Community Health Care Organization, Tokyo Kamata Medical Center.

## Author Contributions

**Conceptualization:** Chihyun Liu, Naoyuki Hayashi.

**Data curation:** Chihyun Liu, Tatsuhiko Kobayashi, Tomoaki Shiba.

**Formal analysis:** Chihyun Liu, Tatsuhiko Kobayashi, Tomoaki Shiba, Naoyuki Hayashi.

**Funding acquisition:** Naoyuki Hayashi.

**Writing – original draft:** Chihyun Liu, Tatsuhiko Kobayashi, Naoyuki Hayashi.

**Writing – review & editing:** Chihyun Liu, Tatsuhiko Kobayashi, Tomoaki Shiba, Naoyuki Hayashi.

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
