## [Decision Letter · Decision Letter 0]

17 Jan 2022

PONE-D-21-25050Effects of aging and exercise habits on blood flow profile of the ocular circulationPLOS ONE

Dear Dr. Hayashi,

Thank you for submitting your manuscript to PLOS ONE. After careful consideration, we feel that it has merit but does not fully meet PLOS ONE’s publication criteria as it currently stands. Therefore, we invite you to submit a revised version of the manuscript that addresses the points raised during the review process.

We look forward to receiving your revised manuscript.

Kind regards,

Barbora Piknova

Academic Editor

PLOS ONE

Journal Requirements:

This work was supported by JSPS KAKENHI Grant Number 21K11643 to NH.

This work was supported by JSPS KAKENHI Grant Number 21K11643.

This work was supported by JSPS KAKENHI Grant Number 21K11643 to NH.

Reviewers' comments:

Reviewer's Responses to Questions

**Comments to the Author**

1. Is the manuscript technically sound, and do the data support the conclusions?

Reviewer #1: No

2. Has the statistical analysis been performed appropriately and rigorously? 

Reviewer #1: No

3. Have the authors made all data underlying the findings in their manuscript fully available?

Reviewer #1: Yes

4. Is the manuscript presented in an intelligible fashion and written in standard English?

Reviewer #1: Yes

5. Review Comments to the Author

Reviewer #1: In this manuscript, the authors report on the effects of exercise on ocular blood flow. They show that several of the markers decrease with age, which is consistent with prior reports. Interestingly, the data suggests that exercise has no beneficial effect on ocular blood flow and, if anything, may be detrimental. This is based on the finding that BOT decreased with age in the exercise group, whereas the no-exercise group did not exhibit the age-related decline. This finding is somewhat paradoxical. While a high number of relatively healthy patients have been recruited for this study and a large number of outcome measures are studied, there are many concerns with the manuscript in its current form. Moreover, due to the lack of detail provided regarding the inclusion/exclusion criteria and questions regarding the statistical analysis, the conclusions aren’t well supported. To the authors credit, they do attempt to address some of the limitations in the discussion.

(1) How are “healthy” people defined in this study. This is a major crux of this study. The inclusion/exclusion criteria and not defined well. Do any of the patients have systemic disease of any type that are treated (or not treated)? Metabolic syndrome is described but there is no data on the percentage of patients with metabolic syndrome nor is it included in the analysis. The lack of adequate patient group matching may have impacted the overall outcome. Sex-linked differences in blood flow suggest hormonal effects. Is the use of oral contraceptives or hormone replacement therapy evaluated?

(2) Patients were excluded if they had an ophthalmic disease. Was a comprehensive eye exam performed? If so, what parameters were evaluated? Was visual acuity measured using EDTRS? Any type of fundus evaluation? OCT of the RNFL? If the presence or absence of ophthalmic disease was based on a medical record, how recent was the medical exam? Were these patients that had all had a recent eye exam within the prior 12 months?

(3) Smoking and alcohol use both affect the vascular system. There is no mention of how alcohol use was quantified. With smoking, current smokers vs non-smokers are included, however prior history of smoking should have also been considered.

(4) Please spell out what MBR stands for.

(5) Was phlebotomy done at the exam? This is assumed but not described. Where was the blood analysis performed? Laboratory specific cut-offs need to be specified so that data can be properly interpreted.

(6) Line 149 states that IOP was measured using a commercial sphygmomanometer. This is incorrect. Please describe how IOP was measured and if repeat measurements were taken. Also of note, was corneal thickness accounted for in the IOP measurement? Thick corneas can artificially inflate pressure readings while thin corneas can give lower readings than actual.

(7) How were the numbers of steps per day calculated? Was this data point only collected for those with exercise watches? How were missed data points accounted for in the statistical analysis?

(8) Numerous questions surround the statistical analysis. It is recommended that a biostatistician review the data in this paper. In the tables, a high number of comparisons are made without accounting for multiple comparisons. For example, in Table 1, there are 31 parameters being compared. It is highly unlikely that a p value of 0.02 in this scenario is relevant. Also, the authors need to address the differences between statistical significance and clinical significance. Many parameters have a p value < 0.05 but have an actual numerical difference of 1 or even less. This isn’t likely to have any clinical meaning.

(9) The term demographics is incorrectly used in table titles. Serum lipid levels or blood pressure are biological metrics but no demographics.

(10) How was waist circumference measured? This is not included in the methods section. How was BMI calculated?

(11) Sex differences in Table 2 are not analyzed.

6. PLOS authors have the option to publish the peer review history of their article (what does this mean?). If published, this will include your full peer review and any attached files.

Reviewer #1: No

---

## [Author Response · Author response to Decision Letter 0]

14 Mar 2022

We thank Reviewer#1 for giving constructive comments and suggesting better phrases. The comments helped us to improve the paper. We stated responses with bullet.

Reviewer #1: In this manuscript, the authors report on the effects of exercise on ocular blood flow. They show that several of the markers decrease with age, which is consistent with prior reports. Interestingly, the data suggests that exercise has no beneficial effect on ocular blood flow and, if anything, may be detrimental. This is based on the finding that BOT decreased with age in the exercise group, whereas the no-exercise group did not exhibit the age-related decline. This finding is somewhat paradoxical. While a high number of relatively healthy patients have been recruited for this study and a large number of outcome measures are studied, there are many concerns with the manuscript in its current form. Moreover, due to the lack of detail provided regarding the inclusion/exclusion criteria and questions regarding the statistical analysis, the conclusions aren’t well supported. To the authors credit, they do attempt to address some of the limitations in the discussion.

(1) How are “healthy” people defined in this study. This is a major crux of this study. The inclusion/exclusion criteria and not defined well. Do any of the patients have systemic disease of any type that are treated (or not treated)? Metabolic syndrome is described but there is no data on the percentage of patients with metabolic syndrome nor is it included in the analysis. The lack of adequate patient group matching may have impacted the overall outcome. Sex-linked differences in blood flow suggest hormonal effects. Is the use of oral contraceptives or hormone replacement therapy evaluated?

>> We added the exclusion criteria in L83 and the data of metabolic syndrome in L93. Metabolic syndrome was not included in the exclusion criteria thus we included them in the analysis.

>> We could not match group since this study used medical checkup data. This may be a task in future study.

>> We did not survey the use of oral contraceptives or hormone replacement therapy. Then we added the effect of them in limitation section (L399).

(2) Patients were excluded if they had an ophthalmic disease. Was a comprehensive eye exam performed? If so, what parameters were evaluated? Was visual acuity measured using EDTRS? Any type of fundus evaluation? OCT of the RNFL? If the presence or absence of ophthalmic disease was based on a medical record, how recent was the medical exam? Were these patients that had all had a recent eye exam within the prior 12 months?

>> Eye diseases was examined by the photograph of the fundus (L89). We do not survey a recent eye exam of the participants. 

(3) Smoking and alcohol use both affect the vascular system. There is no mention of how alcohol use was quantified. With smoking, current smokers vs non-smokers are included, however prior history of smoking should have also been considered.

>> The history of smoking and alcohol use were not surveyed. We added the effect of these on the limitation section (L403).

(4) Please spell out what MBR stands for.

>> Thank the reviewer for pointing this out. We spelled out (L130).

(5) Was phlebotomy done at the exam? This is assumed but not described. Where was the blood analysis performed? Laboratory specific cut-offs need to be specified so that data can be properly interpreted.

>> We added the blood analysis in L155.

(6) Line 149 states that IOP was measured using a commercial sphygmomanometer. This is incorrect. Please describe how IOP was measured and if repeat measurements were taken. Also of note, was corneal thickness accounted for in the IOP measurement? Thick corneas can artificially inflate pressure readings while thin corneas can give lower readings than actual.

>> We edited the measurement of IOP (L159 and L164). 

>> We did not measure corneal thickness this time, and added this point in the limitation section (L415).　

(7) How were the numbers of steps per day calculated? Was this data point only collected for those with exercise watches? How were missed data points accounted for in the statistical analysis?

>> This was collected from the questionnaire (see Suppl.), where participants recorded by themselves. Then recording apparatus was not controlled. We added the limitation of this method in the section (L405).

(8) Numerous questions surround the statistical analysis. It is recommended that a biostatistician review the data in this paper. In the tables, a high number of comparisons are made without accounting for multiple comparisons. For example, in Table 1, there are 31 parameters being compared. It is highly unlikely that a p value of 0.02 in this scenario is relevant. Also, the authors need to address the differences between statistical significance and clinical significance. Many parameters have a p value < 0.05 but have an actual numerical difference of 1 or even less. This isn’t likely to have any clinical meaning.

>> The main purpose of the present study is not to discuss these differences, though there are significant differences in many variables. The fact that effects of age, exercise habits and sex on the blood test results are similar to those of previous studies adds to the reliability of the present study, and these figures may serve as a reference for future studies. Then the statistical tests are left as they are. Whether these differences are clinically meaningful is also not the topic of this study and is thus not discussed.

>> The main purpose of this study was to examine the effects of exercise and aging on ocular flow variables. The effects of aging were similar to previous studies. The absence of significant effect of exercise on ocular flow variables was a new finding of this study.

(9) The term demographics is incorrectly used in table titles. Serum lipid levels or blood pressure are biological metrics but no demographics.

>> The authors edited the title of tables. Thank the reviewer for pointing this out.

(10) How was waist circumference measured? This is not included in the methods section. How was BMI calculated?

We added this point (L161).

(11) Sex differences in Table 2 are not analyzed.

Analysis of sex difference was shown in Table 4.

---

## [Editor Report · Decision Letter 1]

25 Mar 2022

Effects of aging and exercise habits on blood flow profile of the ocular circulation

PONE-D-21-25050R1

Dear Dr. Hayashi,

We’re pleased to inform you that  after the revision your manuscript has been accepted for publication.

Within one week, you’ll receive an e-mail detailing the required technical amendments. When these have been addressed, you’ll receive a formal acceptance letter and your manuscript will be scheduled for publication.

Kind regards,

Barbora Piknova

Academic Editor

PLOS ONE

---

## [Editor Report · Acceptance letter]

6 Apr 2022

PONE-D-21-25050R1 

*Effects of aging and exercise habits on blood flow profile of the ocular circulation*

Dear Dr. Hayashi:

I'm pleased to inform you that your manuscript has been deemed suitable for publication in PLOS ONE. Congratulations! Your manuscript is now with our production department. 

Kind regards, 

on behalf of

Dr. Barbora Piknova 

Academic Editor

PLOS ONE